# The Impact of Systemic Sclerosis on Sexual Health: An Italian Survey

**DOI:** 10.3390/healthcare11162346

**Published:** 2023-08-20

**Authors:** Alessandro Ferdinando Ruffolo, Maurizio Serati, Arianna Casiraghi, Vittoria Benini, Chiara Scancarello, Maria Carmela Di Dedda, Carla Garbagnati, Andrea Braga, Massimo Candiani, Stefano Salvatore

**Affiliations:** 1Obstetrics and Gynecology Unit, IRCCS San Raffaele Hospital, Vita-Salute University, 20132 Milan, Italy; alesruffolo@gmail.com (A.F.R.); casiraghi.arianna@hsr.it (A.C.); benini.vittoria@hsr.it (V.B.); candiani.massimo@hsr.it (M.C.); salvatore.stefano@hsr.it (S.S.); 2Gynecological Department, Jeanne de Flandre Hospital, University Hospital of Lille, Avenue Eugène Avinée, 59037 Lille, France; 3Department of Obstetrics and Gynecology, Del Ponte Hospital, University of Insubria, 21100 Varese, Italy; chiarascanca@gmail.com; 4Department of Obstetrics and Gynecology, ASST FBF-SACCO Macedonio Melloni Hospital, 20129 Milan, Italy; maria.didedda@asst-fbf-sacco.it; 5Fondazione IRCCS Ca’ Granda Ospedale Maggiore Policlinico, University of Milan, Via Commenda 12, 20122 Milan, Italy; carlagarbagnati@mac.com; 6Department of Obstetrics and Gynecology, EOC-Beata Vergine Hospital, 6850 Mendrisio, Switzerland; andrea.braga@eoc.ch

**Keywords:** systemic sclerosis, sexual health, sexual dysfunction, genitourinary syndrome of menopause, vulvovaginal symptoms, quality of life

## Abstract

Objective: To evaluate the impact of systemic sclerosis (SSc) on vulvovaginal atrophy (VVA) and sexual health in an Italian population. Methods: An Italian survey about the prevalence and severity of VVA (on a 0 to 10 scale) and sexual dysfunction (using the Female Sexual Function Index—FSFI) through an anonymous online questionnaire. We investigated couple relationships and intimacy with partners, the predisposition of patients to talk about their sexual problems, physicians’ receptivity, and treatment scenarios. Risk factors for VVA symptoms and sexual dysfunction were assessed. Results: A total of 107 women affected by SSc were enrolled. Of these, 83.2% of women (89/107) complained about VVA symptoms and 89.7% (among sexually active women; 87/97) about sexual dysfunction. Menopausal status did not affect VVA symptoms, while age was the only independent risk factor for sexual dysfunction. About 70% (74/107) of women reported a negative impact of disturbances on intimacy with their partner. A total of 63 women (58.9%) had never discussed their sexual problems and VVA condition with a physician. Lubricants were the only treatment prescribed, and 75% of women would welcome new therapies, even if experimental (62.9%). Conclusions: In women with SSc, VVA symptoms and sexual dysfunction are highly prevalent, independently from menopause. In more than half of the investigated women with SSc, we found reluctance to talk about their sexual problems, despite being symptomatic. This should encourage physicians to investigate vulvovaginal and sexual health. SSc patients would welcome the advent of new treatment possibilities for their VVA and sexual complaints.

## 1. Introduction

Systemic sclerosis (SSc), also known as scleroderma, is an immune-mediated rheumatic disease that causes vasculopathy and fibrosis of the skin and internal organs [1]. SSc has been found to affect various aspects of life, including intimate health and sexuality [2]. Diminished intercourse frequency due to reduction in desire and satisfaction have been related to several SSc symptoms, such as mouth shrinking, skin tightening around vaginal introitus and breast, vaginal dryness, joint pain, muscle weakness, Raynaud phenomenon, reflux, vomiting, diarrhea, low self-esteem, as well as some drugs [3,4,5,6]. Indeed, decreased libido has been related to the use of prednisolone, vaginal ulceration to the use of colchicine, nausea and weakness to the use of nifedipine and prednisolone, and depression to the use of cimetidine [6]. Sexual function has been observed to be significantly impaired, and more sexual distress has been reported in women with SSc in comparison with healthy controls [7] or women affected by other chronic diseases [6]. Several issues, such as intimacy and their relationship with a partner, the spontaneous predisposition of patients to talk about their sexual problems, and physicians’ awareness of or receptivity (meaning the ability to carefully listen to the woman and inquire with interest about possible intimate and relational disturbances) to women affected by SSc, remain largely under-reported [7]. The literature assessing sexuality impairment in women with SSc is growing [3,4,5,6,7,8,9,10]; however, these topics have never been previously explored. 

We, therefore, designed this survey to evaluate the prevalence and severity of vulvovaginal atrophy (VVA) symptoms and sexual health, investigating patients’ relationships with their partners, their physicians’ receptivity, and patients’ satisfaction with the available treatments for their genital and sexual disorders and expectations for alternative therapies in an Italian cohort of women affected by SSc.

## 2. Methods

### 2.1. Study Design

This is an Italian survey concerning the impact of SSc on VVA and sexual health. The questionnaire adopted in this study was designed in collaboration between the Italian National SSc Patient Association, named *Gruppo Italiano per la Lotta alla Sclerodermia *(GILS ODV), the Urogynecology Unit of IRCCS (Istituto di Ricovero e Cura a Carattere Scientifico) at the San Raffaele Hospital of Milan (where data were collected and analyzed), and the following national units: Scleroderma Unit of IRCCS Policlinico of Milan, Scleroderma Unit of ASST (Azienda Socio-Sanitaria Territoriale) Legnano, ASST Niguarda, IRCCS Humanitas Hospital of Milan. In addition to some demographic details and questions designed to investigate patients’ subjective perception of their symptoms and the available solutions, the questionnaire also included the Female Sexual Function Index (FSFI), a validated questionnaire for sexual health assessment (Appendix A). Approval for the study was obtained from the Institutional Review Board (number IRB 20/int/2020). All participants signed informed consent for the treatment of personal data. The study was conducted according to the Declaration of Helsinki [11].

### 2.2. Study Population

Women were enrolled between June 2019 and February 2020 by the previously mentioned Italian units. 

Eligible patients were women with a diagnosis of SSc, according to the American College of Rheumatology [12], over 18 years of age, who signed an informed consent and were willing to fill out the questionnaire. Women who did not sign the informed consent for the treatment of personal data, did not complete the questionnaire, or did not fill in the questionnaire were not included in the study.

### 2.3. Study Procedures

During a routine rheumatologic examination, patients were informed about the possibility of answering anonymously to an online questionnaire based on vulvovaginal atrophy symptoms and sexual health related to systemic sclerosis. 

The questionnaire included questions on general characteristics and medical and surgical history (gynecological surgery, presence or history of malignant neoplasia, frequency of gynecological medical examinations). Menopausal status was defined as “the absence of menstruation for at least 12 months”. Vulvovaginal atrophy symptoms were then assessed through subjective parameters. Vaginal dryness and dyspareunia (defined in the questionnaire as “pain at intercourses”) were investigated using a Visual Analogue Scale (VAS) from 0 to 10 for intensity, where “0” indicated the absence of symptoms and “10” as the maximum intensity. To assess sexual function, we incorporated the Female Sexual Function Index (FSFI), a questionnaire composed of 19 questions to investigate 7 different domains (desire, arousal, lubrification, satisfaction, orgasm, pain, and a total FSFI score). Each domain of the FSFI includes a 0–6 scale where the lower score indicates the worst condition. In order to classify a woman as “sexually dysfunctioned”, the overall FSFI score should be <26.55. Desire is the only item that may be considered separately, with a clinically significant cut-off score for dysfunction <5 [13,14]. FSFI analysis was carried out only for sexually active women, defined as women who had intercourse in the previous 4 weeks. 

Couple relationships and intimacy with partners were then assessed. For this section of the questionnaire, the following questions were proposed to participants: “How much did these symptoms negatively influence your couple relationship and intimacy with partner?”, “Do you think that these symptoms and their influence on sexual life are negatively perceived by your partner?”. We also investigated the spontaneous predisposition of patients to talk about their sexual problems and physicians’ receptivity: “Have you ever discussed about your symptoms (vulvovaginal and sexual) with your doctor?”, “If the answer is yes, which physician did you discussed with?”, “Was the doctor interested and receptive enough on the argument?”, “Who started the conversation about the problem?”.

Previous treatments for VVA complaints and their efficacy were recorded. Finally, we asked each participant if they would consider being submitted to a new or experimental therapy for their VVA symptoms or sexual dysfunction. 

### 2.4. Statistical Analysis

IBM SPSS Statistics for Windows, version 21 (IBM Corp., Armonk, NY, USA) version 21.0 was used to perform data analysis. 

Continuous variables were expressed as mean and standard deviation (SD). Categorical variables were expressed as n (%). Exploratory univariate was applied to test the association between risk factors for VVA-related symptoms and sexual dysfunction. Variables that had a significant association with the adopted scores at univariate analysis were eventually included in the multivariate analyses. A two-tailed *p*-value < 0.05 was considered statistically significant.

## 3. Results

### 3.1. General Characteristics

A total of 133 women were considered eligible for the study and signed the informed consent for the treatment of personal data. However, 19.54% (26/133) dropped out of the study: 53.84% (14/26) did not complete the whole questionnaire (missing data), while 46.16% (12/26) did not fill it at all.

A total of 107 women (80.46%; 107/133), with a mean age of 53.47 (SD ± 13.27) years, affected by SSc were recruited. The clinical and demographic characteristics of the study population are reported in Table 1. About half of the study population was in a menopausal status (58/107; 54.2%), with surgical menopause reported in 9.3% of women (10/107). Most of the included women were sexually active (97/107; 90.7%). 

### 3.2. Vulvovaginal Atrophy (VVA) Symptoms

VVA symptoms are described in Table 2. Vaginal dryness was reported by 83.2% of women (89/107) with a mean severity score of 7.38 (SD ± 1.82), while dyspareunia by 82.2% of women (88/107) with a mean severity score of 7.72 (SD ± 1.44). Age, SSc duration, and menopause did not result related to VVA-related symptoms prevalence.

### 3.3. Sexual Function

In our population, 97/107 (90.70%) women were sexually active. The mean age of sexually active women was 53.28 (SD ± 13.70) years, with SSc duration of 12.76 (SD ± 10.34) years. 

Table 3 shows in detail the FSFI results for a single domain and for the overall score. An FSFI total score < 26.55 was reported in 87/97 (89.7%) of sexually active women. Sexual desire, the only domain that can be considered separately from the FSFI final score, was 2.23 (SD ± 1.00), lower than the cut-off of 5.00 defining the impairment of this FSFI item. While the duration of the SSc and the menopausal status did not reach significance in uni- and multivariate analysis, age was the only condition that resulted as an independent risk factor for sexual dysfunction (Table 4).

### 3.4. Relationship with the Partner

This part of our questionnaire was designed to evaluate the impact of VVA symptoms on partner relationships and intimacy. Only 16/107 (15%) of women reported “no influence” on their intimate relationships, 17/107 (15.9%) reported “little”, 32/107 (29.9%) “fairly”, and 42/107 (39.2%) answered that they were “highly” influenced (Figure 1). Therefore, more than 2/3 of women in our population reported a negative impact on their intimacy because of VVA symptoms. Similarly, the partner’s negative perception related to VVA condition on sexual life was reported by 67/107 (63.2%) women.

### 3.5. Relationship with the Physician

In the study population, 68 (68/107; 63.6%) women had periodical annual gynecological evaluations. Of these, 63 women (63/107; 58.9%) had never discussed their VVA symptoms and the related sexual disorders with their doctor (Figure 2). Overall, 44/107 (41.1%) women revealed their bother to the following physicians: gynecologists in 75% (33/44) of cases, general practitioners in 13.6% (6/44), rheumatologists in 6.8% (3/44), and others in 4.5% (2/44). In the vast majority (40/44; 90.9%), the patient herself reported her symptoms to physicians who, according to the women’s perception, manifested an interest in 84% (37/44) of cases. 

### 3.6. Women’s Considerations of Therapies

Only 15% (16/107) of women were on treatment for their VVA symptoms and, in all cases, with lubricants. Treatment satisfaction was reported as “fairly” in 56.3% (9/16) of cases, as “little” in 31.3% (5/16), and “not satisfied by the treatment” in 12.5% (2/16). The interest in an innovative medication or experimental treatment for their VVA symptoms and sexual dysfunction was expressed by 80/107 (75.5%) of women and 66/107 (62.9%) of cases, respectively.

## 4. Discussion

Chronic rheumatic diseases affect several aspects of women’s life, including sexuality, intimate relationships with partners, and, therefore, their quality of life [15,16].

In women affected by SSc, skin tightening, muscle weakness, joint pain, deformity, and decreased physical function can have a negative impact on female sexuality and sexual functioning [8,17,18]. Common problems reported include vaginal dryness and discomfort, painful intercourse, and reduced frequency and intensity of orgasms [6,19], with a negative impact on intimate relationships and quality of life [20]. 

Our study, conducted on a cohort of 107 women with SSc, detected a VVA symptoms (vaginal dryness and/or dyspareunia) prevalence of 83.2% (89/107). The prevalence observed was greater than that reported in previous studies [6,21], with a severe perceived intensity of vaginal dryness and dyspareunia (7.38 and 7.72 on a 0 to 10 scale, respectively). As a consequence of VVA symptoms, sexual function was greatly impaired, too. Despite the high rate of sexually active women (90.7%; 97/107) in our population, 89.7% (87/97) of them were affected by sexual dysfunction, reporting low scores in all sexual domains, with a mean FSFI total score of 15.66. Moreover, even sexual desire, the only item that can be considered separate from the FSFI total score, was highly impacted with a mean of 2.23, far lower than the general cut-off of 5.00. In our analysis, the sexual dysfunction rate was higher than other rates reported in previous studies, while the intensity of sexual impairment evaluated with the FSFI total score was in line with the literature [20,21,22]. 

Levis et al. [9] led a study on 165 sexually active women affected by SSc and complaining of sexual impairment, reporting a 61.8% sexual dysfunction rate. The authors, however, adopted an FSFI total score cut-off of 22.5, lower than the cut-off of 26.55 commonly used, and included only a few women in menopause (less than 1/3 of the study population). Shouffer et al. [7] reported a sexual dysfunction rate of 70% in a population of 37 women affected by SSc, with only 27% of women in menopause. Menopause is known to be one of the main factors determining sexual dysfunction as a part of the genitourinary syndrome of menopause (GSM). Tissue changes in the external and internal female genitalia include retraction of the introitus, thinning and regression of the labia, and prominence of the urethral meatus [23], similar to changes reported in women affected by SSc [3,4,5,6], leading to VVA symptoms such as vaginal pain, dyspareunia, dryness, itching, and tissue friability. However, in our cohort of women affected by SSc, the disease itself seems to be a fundamental factor impacting VVA symptoms prevalence and sexuality, regardless of the menopausal status. Different authors have reported a worsening of sexual function impairment with age in women with SSc [6,7,8,9,10,18,19,20,21,22,24]. Our data demonstrated that SSc highly impacts VVA symptom prevalence and sexual dysfunction. In our study, when the risk factors for VVA symptoms and sexual dysfunction were assessed, patients’ age resulted as the only risk factor related to sexual dysfunction. Indeed, while age, SSc duration, and menopause did not show to affect VVA symptom prevalence, older age was related to a higher sexual dysfunction prevalence. These findings led us to conclude that sexuality is affected independently by SSc but worsened by age. 

Furthermore, the assessment of patients’ relationships with their partners, communication with their physicians, and the desire for a novel outpatient treatment for genital symptoms was of primary importance in our survey. 

Our data show that 70% of women affected by SSc felt a negative impact of VVA symptoms and impaired sexual function in intimacy with their partner and that in almost two-thirds of cases, these symptoms and their influence on sexual life are negatively perceived by the partner. Unfortunately, all these personal aspects still represent a taboo: sexual health is rarely investigated by physicians, and less than half of our patients (41.1%) search for a medical opinion. When a physician is involved by women in the assessment of VVA symptoms and sexual health, our survey reported gynecologists as the reference figure (30.84%). However, this study population, in which more than half of women have an annual gynecological examination, is only a part of a larger population in which the gynecologist could not be the reference figure. Therefore, it is possible that in other institutions, VVA symptoms and sexual problems are even less discussed by women to their own medical doctors. Indeed, while Schouffoer et al. recommended a routine interest of the physician about sexual problems in SSc women [16], Knafo et al. criticized this wide approach, finding that only a few patients desire to discuss sexual problems, with an infrequent involvement of rheumatologists [25], and suggesting, therefore, a routine assessment of sexual function only in the case of an effective intervention and patient benefit. In our opinion, educational activities on intimate aspects are needed for women with SSc in order to improve their awareness about this condition and the research of effective treatments. Educational activities about VVA symptoms and sexual disorders in women affected by SSc should be encouraged with two main targets: the patients and the medical caregivers. Firstly, women affected by SSc should be aware that VVA symptoms and sexual problems may be more frequent than in the healthy female population. Women affected by SSc should seek medical advice in case of vulvovaginal symptoms and sexual impairment, considering that appropriate treatments and strategies for this condition are available. Moreover, educational meetings should be encouraged through Patient Associations, with the aim of spreading awareness about this aspect of SSc among patients. On the other hand, educational activities should directly concern medical caregivers. Indeed, physicians assisting women with SSc should be systematically educated and trained in the screening and management of this important complication of SSc, which, even if it does not result in a decreased life expectancy, has a very high incidence with a huge impact on women’s quality of life and health.

Concerning treatments, only a small part of the included women (16%) have adopted just lubricants for their symptoms without reaching satisfactory efficacy. Indeed, the majority of the study population would try a new (75.5%) or experimental (62.9%) treatment for vulvovaginal atrophy and sexual dysfunction treatment.

Several therapies are already available to treat VVA symptoms in menopausal women [26,27], including both systemic therapies such as ospemiphene [28], and local therapies such as vaginal estrogens and vaginal laser energies like the microablative fractional CO_2_ laser or the erbium YAG laser [29,30], but none of these has been investigated in women affected by SSc. Further studies, however, are needed to evaluate the safety profile and the efficacy of these treatments on postmenopausal women affected by SSc complaining of VVA symptoms and sexual dysfunction.

The main strength of the current survey is the assessment of sexual health with a questionnaire carried out in collaboration with the National Italian Association of Scleroderma Patients named GILS ODV (*Gruppo Italiano Lotta alla Sclerosi Sistemica*) on the basis of the most frequent dilemma reported by SSc women, in association with a validated questionnaire (the FSFI). Moreover, the adoption of an anonymous questionnaire filled independently aimed to shield the result from the embarrassment provoked by the topic. Questionnaires, moreover, represent a feasible, non-invasive, and non-expensive method of investigation, making our study particularly acceptable to our patients. 

Even if the sample size of 107 women with SSc can affect the statistical models, such as the multivariate analysis, we consider this population size as another strength of our investigation. SSc is, in fact, a rare medical condition with a prevalence that falls between 38 and 341 cases per million persons [31]. When assessing sexual function, our sample size of 107 women is one of the largest reported in the literature. 

Furthermore, we deeply investigated new scenarios of SSc patients; we discovered an important negative influence on couple relationships due to sexual dysfunction, and we noticed communication difficulties with physicians. We highlighted the need for new effective therapies for this condition, as evidenced by the majority of SSc women. For vulvovaginal atrophy, therapeutic alternatives have increasingly been developed that result in improving symptoms and are also safe in terms of side effects and contraindications. Mechanical therapies, in particular, the vaginal CO_2_ laser, are proving to be effective and safe in both clinical and histopathological terms [30]. The remodeling effect of the vaginal laser, which consists of the modification and regeneration of the vaginal mucosa through an increase in the content of collagen and elastic fibers, could be useful for women affected by SSc, considering the type of pathophysiology that sustains the tissue modification in affected women. In addition to that, in patients who are already using pharmacological therapies for their condition and have several comorbidities, offering a treatment that does not have severe side effects or pharmacological interactions may be of fundamental importance. Pilot studies and randomized control trials assessing the safety and efficacy of the different treatment strategies would be desirable in order to properly suggest adequate therapies for this condition. 

However, this study presents several limitations, such as the impossibility of matching the results of the questionnaires to the clinical conditions related to SSc, because of the anonymous format of the questionnaire. Moreover, surveys may be affected by some typologies of bias intrinsic in such type of design. Indeed, considering the online design of the survey, participants may have answered questions inaccurately (response and recall bias). Moreover, wording differences can confuse the respondent or lead to incorrect interpretations of the question, especially for non-validated questions (question-wording). Furthermore, considering that 90.7% of the study population was sexually active, women who agreed to be enrolled in the study and completed the online questionnaire may be more interested in genital and sexual health than the general population of women affected by SSc (selection bias/sampling frame), making impossible an absolute generalizability to all patients with SSc.

## 5. Conclusions

Women with SSc present a high prevalence and severity of VVA symptoms and sexual dysfunction. These findings seem to be independent of the menopausal status and SSc duration. However, in a population of women affected by SSc, elderly patients are more frequently affected by sexual dysfunction. Relationships with their partners are highly impaired in this category of women. This increased awareness concerning the development of VVA and sexual dysfunction in women with SSc should help physicians investigate aspects that patients are partially reluctant to reveal spontaneously. Considering VVA management, not only lubricants should be considered as a possible treatment since patient satisfaction is not optimal. Research should be carried out in this field to explore new treatment strategies that would be welcome by women themselves. 

## Figures and Tables

**Figure 1 healthcare-11-02346-f001:**
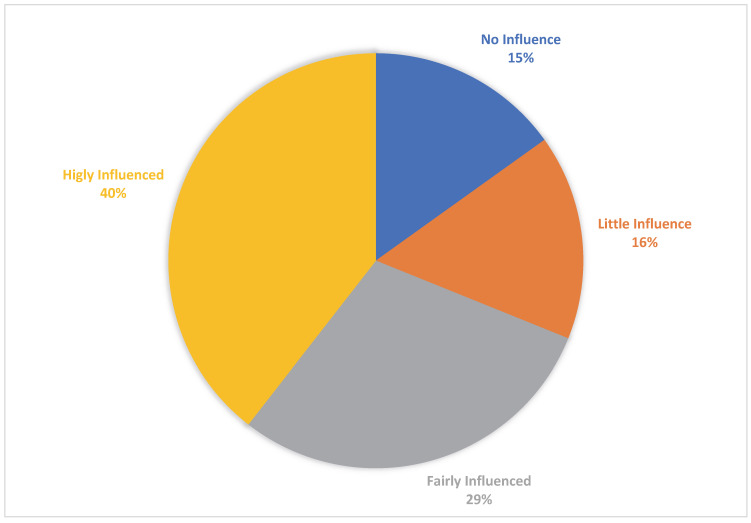
Influence of systemic sclerosis on intimacy with their partners reported by affected women.

**Figure 2 healthcare-11-02346-f002:**
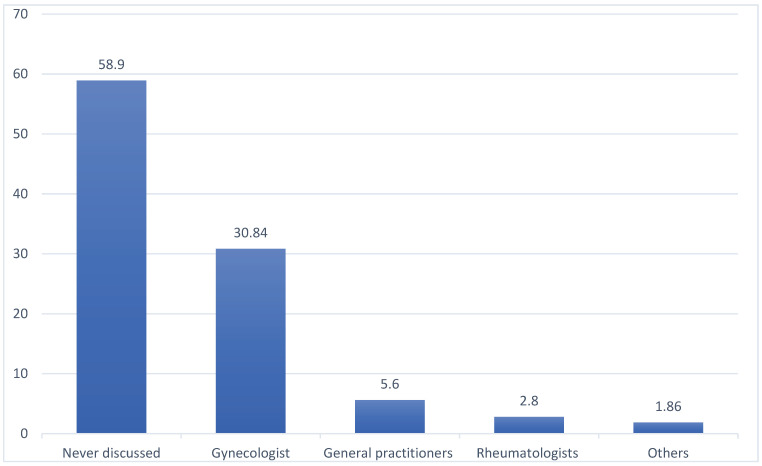
Kind of physicians involved in the assessment of women’s vulvovaginal atrophy symptoms and sexual health.

**Table 1 healthcare-11-02346-t001:** Demographic and clinical characteristics of the study population. SD: Standard Deviation.

	N = 107
Age, mean (±SD), years	53.47 (±13.27)
Systemic sclerosis duration, mean (±SD), years	12.48 (±10.28)
Menopausal status, n (%)	58 (54.2)
Hysterectomy, n (%)	13 (12.1)
Bilateral salpingo-oophorectomy, n (%)	10 (9.3)
Malignant neoplasia, n (%)	3 (2.8)
Sexually active women, n (%)	97 (90.7)
Age of sexually active women, mean (±SD), years	53.28 (±13.70)
Systemic sclerosis duration of sexually active women, mean (±SD), years	12.76 (±10.34)

**Table 2 healthcare-11-02346-t002:** Vulvovaginal atrophy symptoms of the study population. ^§^ Calculated among women complaining of vaginal dryness. * Calculated among women complaining of dyspareunia.

	N = 107
Vaginal dryness, n (%)	89 (83.2)
Vaginal dryness severity ^§^, mean (±SD)	7.38 (±1.82)
Dyspareunia, n (%)	88 (82.2)
Dyspareunia severity *, mean (±SD)	7.72 (±1.44)

**Table 3 healthcare-11-02346-t003:** Sexual function evaluated by the Female Sexual Function Index of the study population. Calculated among sexually active women. SD: Standard Deviation.

	N = 97
Sexual dysfunction, n (%)	87 (89.7)
Desire, mean (±SD)	2.23 (±1.00)
Arousal, mean (±SD)	2.66 (±1.74)
Lubrification, mean (±SD)	2.56 (±1.91)
Orgasm, mean (±SD)	2.69 (±2.04)
Satisfaction, mean (±SD)	2.93 (±1.84)
Pain, mean (±SD)	2.57 (±2.05)
Total, mean (±SD)	15.66 (±9.36)

**Table 4 healthcare-11-02346-t004:** Factors related to vulvovaginal atrophy-related symptoms and sexual dysfunction. * Calculated among sexually active women. HR: Hazard Ratio; CI: Confidence Interval.

	Univariate		Multivariate	
	**HR (95%CI)**	***p*-Value**	**HR (95%CI)**	***p*-Value**
	**Vulvovaginal Atrophy Symptoms**			
Age, years	1.05 (1.01–1.13)	**0.01**	1.04 (0.98–1.11)	0.17
Systemic sclerosis duration, years	1.06 (0.99–1.13)	0.09		
Menopause	3.43 (1.11–10.58)	**0.03**	1.44 (0.27–7.61)	0.66
	**Sexual Dysfunction ***			
	**HR (95%CI)**	***p*-Value**	**HR (95%CI)**	***p*-Value**
Age, years	1.08 (1.02–1.15)	**0.002**	1.08 (1.00–1.18)	**0.04**
Systemic sclerosis duration, years	1.01 (0.95–1.02)	0.61		
Menopause	5.16 (1.03–25.71)	**0.02**	0.90 (0.08–9.38)	0.93

Bold numbers evidence a statistical significance (*p* < 0.05).

## Data Availability

Data are available on request from the authors. The data that support the findings of this study are available from the corresponding author, AFR, upon reasonable request.

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
