# Peer review of "The Impact of Systemic Sclerosis on Sexual Health: An Italian Survey"

_healthcare, 2023, doi:10.3390/healthcare11162346_

Round 1
Reviewer 1 Report
The manuscript contain very important information in the area of SSc, however, the presentation of results needs to be improved. There are tables following each other in the document. These tables should be separated by text. I have also noted that the table caption is at the bottom of the table. The usual way is that table captions are placed on top of the table, while figure caption is placed at the bottom.
There is also a need to proof read the whole manuscript and correct the English grammar, eg Lines 145, 194, 196, 198, 199 have problems with the word "resulted" and 220 ...the higher is the age......needs to be corrected.
On results, age was the only significant predictor in the multivariate model. The sample size of 103 may have an effect on the multivariate model. However, since the SSc is a rare condition, it is worthy mentioning about the sample size in relation to the multivariable analysis.
Otherwise this is a good manuscript that should be published after incorporating the comments.
The quality of English language in the manuscript needs to be improved. There is a need to proof read the whole manuscript and correct the English grammar, eg Lines 145, 194, 196, 198, 199 have problems with the word "resulted" and 220 ...the higher is the age......needs to be corrected.
Author Response
The manuscript contain very important information in the area of SSc, however, the presentation of results needs to be improved. There are tables following each other in the document. These tables should be separated by text and table captions are placed on top of the table. The usual way is that table captions are placed on top of the table, while figure caption is placed at the bottom.
Thank you for your revision and your time. We’ll try to answer to your concerns at our best.
R- The captions are now placed below the tables, which are all separated by text, except for table 3 and 4 that refer to two features of the same paragraph.
There is also a need to proof read the whole manuscript and correct the English grammar, eg Lines 145, 194, 196, 198, 199 have problems with the word "resulted" and 220 ...the higher is the age......needs to be corrected.
R- We corrected the manuscript as you indicated.
On results, age was the only significant predictor in the multivariate model. The sample size of 103 may have an effect on the multivariate model. However, since the SSc is a rare condition, it is worthy mentioning about the sample size in relation to the multivariable analysis.
R-We will mention the effect of the sample size on the multivariate analysis model in the limitations of the study.
Reviewer 2 Report
Thank you for submitting your manuscript. I really enjoyed learning about the relationship between these two disorders. Here are my comments below to help improve the quality of the manuscript.
Abstract
1. Line 22- I do not think the term multicentric is the term that can describe the survey.
2. Line 21- I am not sure what you mean by Vuvlovaginal symptoms? Do you mean Vulvovaginal Candidiasis? Or Vaginal Atrophy? or Vulvovaginitis?
Line 30- you can put the number 63.
Introduction
1. You defined what SSc was but failed to define what vulovvaginal symptoms are. There is also issues or disagreements with claiming female sexual dysfunction and whether this is a true phenomenon.
Methods
Did you pilot the new survey? You used, what I assume, was a validated survey but then changed/ modified it. How were these sexual health questions validated? Also, mention of the use of FSFI should be mentioned in your study design.
Study Population- I would have liked to see a breakdown of demographics k\(age, education, etc).
In your analysis section, you discussed running a univariate, but you also did a multivariate analysis. This is missing in your methods section.
Results
Your tables are hard to read- you have SD and % (Frequency) in the same table. It is hard to keep tracking if you are discussing the SD or % in the tables. You could mention the SD's in your results and only use N(%) in your tables.
For Relationship with Partner- was the question does VV impact your intimacy with your partner or something like that? The discussion reads as if the question was "does your partner influence your intimacy?".
Author Response
Thank you for submitting your manuscript. I really enjoyed learning about the relationship between these two disorders. Here are my comments below to help improve the quality of the manuscript.
Thank you for your time and your precious comments. We wish to have properly answered to your questions.
Abstract
Line 22- I do not think the term multicentric is the term that can describe the survey.
R- We defined the trial multicentric due to the fact that the questionnaire was submitted to patients referred to scleroderma specialized centers in different institutions in northern Italy and also in collaboration with the Italian Patient Association “Gruppo Italiano per la Lotta alla Sclerodermia (GILS)”, that collects patients’ membership nationally. However, we can remove this term from the methods paragraph.
- Line 21- I am not sure what you mean by Vuvlovaginal symptoms? Do you mean Vulvovaginal Candidiasis? Or Vaginal Atrophy? or Vulvovaginitis?
R- For VV symptoms we meant symptoms related to vulvovaginal atrophy, so vaginal dryness and dyspareunia, as discussed in methods after. We’ll try to give homogeneity to the manuscript with this definition.
Line 30- you can put the number 63.
R- corrected
Introduction
You defined what SSc was but failed to define what vulovvaginal symptoms are. There is also issues or disagreements with claiming female sexual dysfunction and whether this is a true phenomenon.
R-We re-defined vulvovaginal symptoms as vulvovaginal atrophy symptoms in the manuscript. Female sexual dysfunction was assessed through a validated Italian questionnaire, the Female Sexual Function Index, that has been designed with the aim of detect sexual dysfunction and so confirming the real presence of this condition.
Methods
Did you pilot the new survey? You used, what I assume, was a validated survey but then changed/ modified it. How were these sexual health questions validated? Also, mention of the use of FSFI should be mentioned in your study design.
R- We used the FSFI that has already been validated and widely used for the assessment of sexual health. To this we added questions regarding the patient's personal experience of her complaints and her experience in communicating these complaints with the physician figure. The questionnaire was collaboratively developed based on the clinical experience and issues most often encountered in the daily lives of caregivers who follow these patients and with the help of the Italian SSc Patient Association (GILS), as mentioned in the Methods paragraph. As you suggested, we already mentioned the FSFI in the study design too.
I would have liked to see a breakdown of demographics k\ (age, education, etc). In your analysis section, you discussed running a univariate, but you also did a multivariate analysis. This is missing in your methods section.
R- Since we designed this online survey with the assessment of several issues not previously evaluated, such as the patients’ relationship with their partners, physicians’ receptivity and patients’ satisfaction with the available treatments for their VVA and sexual disorders and expectations for alternative therapies, we thought our questionnaire to be already full of questions, and we did not want to lose patients for a decreased compliance. So, unfortunately, all the available baseline characteristics of the study population are already reported in table 1.
Results
Your tables are hard to read- you have SD and % (Frequency) in the same table. It is hard to keep tracking if you are discussing the SD or % in the tables. You could mention the SD's in your results and only use N(%) in your tables.
R- We established the use of SD and % in the first column. I’m sorry if reading this table is hard or difficult, but the baseline characteristics of the study population are frequently reported like that in several manuscripts, and we think that dividing these data between the text (SD) and tables (N%) would make it really confused and harder to interpret.
For Relationship with Partner was the question does VV impact your intimacy with your partner or something like that? The discussion reads as if the question was "does your partner influence your intimacy?".
R- In methods we reported the questions we administered to women in the survey: Couple relationship and intimacy with partner were then assessed. For this section of the questionnaire, the following questions were proposed to participants: “How much did these symptoms negatively influence your couple relationship and intimacy with partner?”, “Do you think that these symptoms and their influence on sexual life are negatively perceived by your partner?”. The first question clearly aims to evaluate the intimacy perception of the woman affected by SSc, while the second one if the also the partner is influence by the SSc condition.
In the results we stated: Only 16/107 (15%) of women reported “no influence” on their intimate relationships, 17/107 (15.9%) reported “little”, 32/107 (29.9%) “fairly” and 42/107 (39.2%) answered that they were “highly” influenced (Figure 1). Therefore, more than 2/3 of women in our population had a bad impact on their intimacy because of VVA symptoms. Similarly, the partner’s negative perception related to VVA condition on sexual life was reported by 67/107 (63.2%) women. So, we reported at the beginning the patient perception, and at the end the partner perception.
Finally in the discussion we reported: Our data show that about 70% of women have reported a negative impact of VVA symptoms and impaired sexual function in the intimacy, with often a reduced quality of sexual life perceived by partners. It does not seem to us that the discussion reads as you reported.
However, we can follow your suggestion and try to rewrite this concept in order to make it clearer.
Reviewer 3 Report
This is an interesting manuscript that aimed to evaluate the impact of systemic sclerosis on vulvovaginal symptoms and sexual health in an Italian population. However, before the paper can be accepted in the journal Healthcare, a thorough revision has to be made.
Major issues:
Was the survey used in the study validated? If so, can you reference previous uses of the study in Methods section? If not, is there some data on reliability/validity?
More study limitations should be emphasized, except for selection bias and inability to draw clinical conclusions. What about response bias, recall bias, limited generalizability, measurement errors, question wording, sampling frame, etc. Please mention and expand those you believe are relevant.
The Abstract should be restructured. More emphasis should be placed on the methods, and the conclusion is not clear in how it is written (for example, phrases such as "our population" should be avoided).
Minor issues:
The manuscript as a whole should be proofread by a native English speaker, as there are many typographical errors and erroneous sentence constructions.
Some group of drugs that are responsible for diminished intercourse frequency should be mentioned, based on the literature (line 47)
Please explain better the syntagm "physicians' awareness or receptivity" (lines 51-52)
Please highlight and write the number of the Approval from the Institutional Review Boards
Declaration of Helsinki is written erroneously; also, it could be referenced
Please explain all abbreviations when initially mentioned; the reader is not familiarized with IRCSS, ASST, etc.
SPSS should be cited properly - IBM SPSS Statistics for Windows, version 21 (IBM Corp., Armonk, N.Y., USA)
The authors state "A p-value < 0.05 was considered statistically significant", but it should be stated whether it was two-tailed.
The authors emphasize "In our opinion, educational activities on intimate 239 aspects are needed in women with SSc, in order to improve the awareness about this condition and further treatments" (lines 239-241). How would these educational activities be developed and implemented?
The authors state that they have "highlighted the need for new effective therapy for this condition"; can you please expand on this?
In the Conclusion section, lubricants are mentioned as treatment; are they actually a management strategy?
Many references are not in line with journal instructions, this should be amended.
The manuscript as a whole should be proofread by a native English speaker, as there are many typographical errors and erroneous sentence constructions.
Author Response
This is an interesting manuscript that aimed to evaluate the impact of systemic sclerosis on vulvovaginal symptoms and sexual health in an Italian population. However, before the paper can be accepted in the journal Healthcare, a thorough revision has to be made.
Thank you for your revision and your time. We really appreciated your suggestions and we hope to have well answered to your concerns.
Major issues:
Was the survey used in the study validated? If so, can you reference previous uses of the study in Methods section? If not, is there some data on reliability/validity?
R- As reported in the methods, this survey was designed in collaboration with the Italian National SSc Patient Association, named Gruppo Italiano per la Lotta alla Sclerodermia (GILS ODV), and the medical doctors with high expertise in this field. Several issues concerning different unexplored domains in these kind of patients, such as the patients’ relationship with their partners, physicians’ receptivity and patients’ satisfaction with the available treatments for their VVA and sexual disorders and expectations for alternative therapies, were adopted for the study purpose. Concerning the evaluation of clinical subjective aspects, a validated questionnaire (the Female Sexual Function Index-FSFI), was adopted to assess sexual health.
More study limitations should be emphasized, except for selection bias and inability to draw clinical conclusions. What about response bias, recall bias, limited generalizability, measurement errors, question wording, sampling frame, etc. Please mention and expand those you believe are relevant.
R- We agree with your consideration. All the bias you listed are intrinsic in a survey evaluation and may have interfered with the study results. We have no problem in list them in the limitations section as you suggested.
The Abstract should be restructured. More emphasis should be placed on the methods, and the conclusion is not clear in how it is written (for example, phrases such as "our population" should be avoided).
R- World count limitations is frequently an enemy of a well-written abstract. We tried to fix it according to your suggestions.
Minor issues:
Some group of drugs that are responsible for diminished intercourse frequency should be mentioned, based on the literature (line 47)
R- we reported these drugs in the manuscript.
Please explain better the syntagm "physicians' awareness or receptivity" (lines 51-52)
R- we added details of what we meant.
Please highlight and write the number of the Approval from the Institutional Review Boards
R- IRB 20/int/2020
Declaration of Helsinki is written erroneously; also, it could be referenced
R- corrected
Please explain all abbreviations when initially mentioned; the reader is not familiarized with IRCSS, ASST etc.
R-corrected
SPSS should be cited properly - IBM SPSS Statistics for Windows, version 21 (IBM Corp., Armonk, N.Y., USA)
R- corrected
The authors state "A p-value < 0.05 was considered statistically significant", but it should be stated whether it was two-tailed.
R- We stated it in the manuscript.
The authors emphasize "In our opinion, educational activities on intimate 239 aspects are needed in women with SSc, in order to improve the awareness about this condition and further treatments" (lines 239-241). How would these educational activities be developed and implemented?
R- thank you very much. We implemented this important topic in the manuscript.
The authors state that they have "highlighted the need for new effective therapy for this condition"; can you please expand on this?
R- thank you, we expanded the concept in the manuscript.
In the Conclusion section, lubricants are mentioned as treatment; are they actually a management strategy?
R- Yes, they are considered the first-line treatment for vaginal atrophy.
Many references are not in line with journal instructions, this should be amended.
R– we reviewed references, and we corrected what we could. It should be highlighted that several references come from books, differing from articles citations, as you well know. However, if you have precise indications, we’ll be happy to correct them.
Round 2
Reviewer 2 Report
Thank you for making the corrections that the reviewers have suggested.
Reviewer 3 Report
Thank you for the modification; the article can now be accepted in the current form.
Minor changes needed.